# Integrating User Feedback under Identity Uncertainty in Knowledge Base Construction

**Ari Kobren**                                                                          AKOBREN@CS.UMASS.EDU
**Nicholas Monath**                                                                  NMONATH@CS.UMASS.EDU
**Andrew McCallum**                                                                MCCALLUM@CS.UMASS.EDU
*University of Massachusetts Amherst*
*College of Information and Computer Sciences*
*140 Governors Drive,*
*Amherst, MA 01002 USA*

## Abstract

Users have tremendous potential to aid in the construction and maintenance of knowledges bases (KBs) through the contribution of feedback that identifies incorrect and missing entity attributes and relations. However, as new data is added to the KB, the KB entities, which are constructed by running entity resolution (ER), can change, rendering the intended targets of user feedback unknown–a problem stemming from *identity uncertainty*. In this work, we present a framework for integrating user feedback into KBs in the presence of identity uncertainty. Our approach is based on representing user feedback as mentions and then reasoning jointly about feedback and mentions during ER. We propose a specific representation of user feedback as *feedback mentions* and introduce a new online algorithm for integrating these mentions into an existing KB. In experiments, we demonstrate that our proposed approach outperforms the baselines in 70% of experimental conditions.

## 1. Introduction

Structured knowledge bases (KBs) of entities and relations are often incomplete and noisy, whether constructed by hand or automatically. For example, it has been reported that 71% of people in Freebase are missing a place of birth attribute and 75% have no known nationality [Dong et al., 2014a]. Similarly, while YAGO2 is estimated to be about 95% accurate on facts extracted from Wikipedia, this translates to roughly 5.7 million incorrect facts involving 2.6 million entities[1] [Hoffart et al., 2013]. The vast research in cleaning and correction of databases is further evidence of the permeation of errors throughout KB construction in multiple domains [Dong et al., 2014a,b, Wang et al., 2015, Li et al., 2017].

As the primary consumers of KBs, human users have significant potential to aid in KB construction and maintenance. From a user's standpoint, a KB contains a set of entities, each entity possessing attributes and optionally participating in relationships with other entities. Thus, KB errors manifest as spurious and missing attributes and relationships. However, the data that gives rise to a KB is a collection of raw evidence, which can be understood as *mentions* that require clustering by entity resolution (ER) into a set of *inferred entities*. The attributes and relations of the inferred KB entities with which the user interacts are drawn from this underlying clustering of the mentions. Therefore, the

---

1. Calculated from Table 5 of Hoffart et al. [2013]: 124,333,521 facts with 95.4% weighted average accuracy across relations.

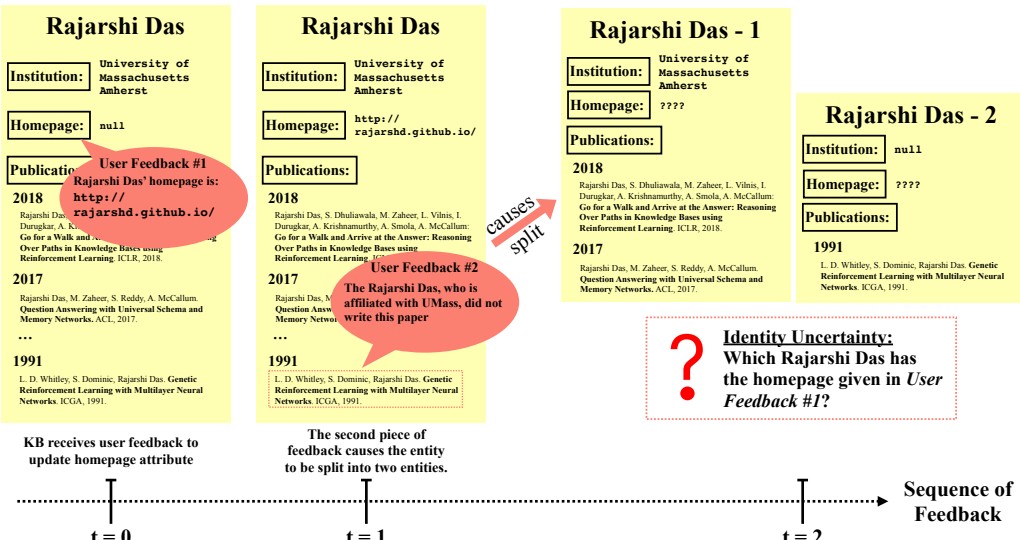

Figure 1: **Example of identity uncertainty inherent in user feedback.** The figure represents the state of the KB for the entity, *Rajarshi Das*, before and after two pieces of feedback are given. *User Feedback #1*, which provides the homepage attribute, could refer to either *Rajarshi Das-1* or *-2*, neither of which existed at the time the feedback was given.

spurious and missing attributes and relationships, may stem from a variety of sources, including: noisy mentions produced by information extraction, mistakes in ER, missing data, etc.

In light of new data that is continually being added to a KB, inferred KB entities may change. Specifically, the arrival of new mentions and user feedback can trigger modifications of the underlying mention clustering, resulting in the creation of new inferred entities, removal of previously inferred entities or alteration of the existing inferred entities' attributes and relations. The volatility of the underlying mention clustering poses a formidable challenge to the task of integrating user feedback with KB content, especially when the precise targets of feedback are unknown, a phenomenon known as *identity uncertainty* [McCallum and Wellner, 2003, Pasula et al., 2003].

As an example, consider Figure 1, which displays an entity in a KB of researchers and user feedback provided about that entity. First, a user notices the entity `Rajarshi Das` is missing a homepage and so provides `rajarshd.github.io`. Later, a user provides feedback that the paper, which was published in 1991 and titled *Genetic Reinforcement Learning with Multilayer Neural Networks*, was not written by the `Rajarshi Das` affiliated with *University of Massachusetts Amherst*. This feedback causes the `Rajarshi Das` entity to be split into two entities. After the split, it cannot be determined which of the two newly created entities should have the homepage provided by the first piece of user feedback. The uncertainty arises from the ambiguity of which true identity of Rajarshi Das is referred to in the user feedback providing the homepage.

In this paper, we present a new framework for reasoning about user feedback amidst identity uncertainty (§3, §4). Our approach is founded on the idea that user feedback should

participate in ER alongside standard mentions. That is, when an inferred entity that was previously the target of user feedback is split–as in the example above–the decision about the new target of the feedback should be made by the ER algorithm. Including feedback in ER, rather than applying feedback to the inferred entities as a post-hoc step, is powerful: the feedback provides additional evidence for mergers and splits, making ER more robust and effective.

A direct way to enable this type of reasoning about user feedback is to *encode the feedback as mentions* and run ER on the union of feedback and standard mentions. However, this approach may result in additional ER errors. To see why, consider the second piece of feedback in the example above. Before the feedback is integrated, it is incompatible with `Rajarshi Das`, since the KB entity exhibits a publication record (the paper from 1991) that the feedback explicitly does not. Ideally, ER would associate the feedback with `Rajarshi Das`, despite the inconsistency, and use the conflicting publications (and other information such as affiliation) as evidence in support of splitting of the KB entity's underlying mention cluster. To enable this, we advocate for the representation of user feedback as *feedback mentions* (FMs), which include two components: a *packaging* (§3.1) and a *payload* (§3.4). The packaging contains attributes used by ER to initially determine a set of mentions similar to the feedback. After this initial set is discovered, attributes contained in the payload are applied, introducing missing attributes, correcting spurious attribute and influencing future ER decisions. In the example above, a FM with packaging that includes an affiliation with *University of Massachusetts Amherst* helps to guide ER toward associating the feedback with the mentions that constitute KB entity `Rajarshi Das`. Afterward, the FM's payload, which includes the incompatibility with published paper from 1991, helps ER to correctly split the mentions into two inferred entities.

User feedback and new mentions naturally arrive over time, rather than all at once, thereby demanding an online integration scheme. In this work, we also present a new incremental ER algorithm for integrating new mentions and FMs with an existing KB (§4). Our algorithm builds a hierarchical clustering of the data, one data point at a time. The crux of the algorithm is a `graft` subroutine, which allows new data to trigger cascading mergers of disparate parts of the tree to produce new and split existing inferred entities.

Previous work studies user feedback for ER, but is focused on user supplied, mention-level pairwise constraints, which are insufficient for providing feedback about KB entities, their attributes and relations. For example, using pairwise constraints alone, it is impossible to supply missing attributes and relations or correct for noise in the underlying mentions. Similarly, many types of desirable user feedback–as in the example above–are inexpressible in the language of pairwise constraints. Because the number of possible pairwise constraints is large, collecting pairwise feedback introduces an additional challenge of designing specialized strategies for selecting which pairs to label and in what order [Wang et al., 2012, 2013, Firmani et al., 2016].

Although user feedback can be used to correct arbitrary KB errors, in this paper we focus on leveraging user feedback for correcting mistakes in ER. We conduct two experiments (§6) in the context of *author disambiguation*–an instance of ER commonly studied for building KBs of researchers. In the first, we automatically generate user feedback that includes the areas of expertise, represented as a set of keywords, for various authors (§6.4). In the second experiment, we generate user feedback that identifies missing and incorrectly

attributed publications with respect to a set of currently inferred KB entities (§6.5). We propose three baseline approaches for integrating user feedback and two feedback simulation schemes and measure the number of pieces of feedback required to recover the ground-truth entities under each experimental setting. Our results show that our proposed approach based on FMs outperform the baselines in 70% of experimental conditions. Our work initiates the investigation of user feedback integration amidst identity uncertainty in KBs, an under-explored and important problem whose solution would dramatically improve the effectiveness of users in the process of KB construction and maintenance.

## 2. Entity Resolution for KB Construction

Our goal is to construct a framework for automatically reasoning about the integration of user feedback and KBs under identity uncertainty. In this section, we define formal models for mentions and entities, which serve as building blocks for the remainder of our discussion.

### 2.1 Mentions, Entities and Attributes

A KB is comprised of a set of mentions $\mathcal{M} = \{x_0, \cdots, x_n\}$ which refer to a set of ground truth entities $\mathcal{E}^\star = \{e_0^\star, \cdots, e_k^\star\}$. Each mention, $x_i$, refers to exactly one ground-truth entity, denoted $e^\star(x_i)$. The goal in ER is to construct a partition of the mentions, $\hat{\mathcal{E}} = \{\hat{e}_0, \cdots, \hat{e}_l\}$ as similar to $\mathcal{E}^\star$ as possible. Each $\hat{e} \in \hat{\mathcal{E}}$ is known as an *inferred entity*.

Mentions are comprised of *attributes*, which serve as evidence of inferred entity attributes and relations. Each mention, $x_i \in \mathcal{M}$, has a corresponding set of attributes, $\mathcal{A}(x_i) = \{a_0, \cdots, a_m\}$, which is a subset of the entire set of attributes $A$, i.e., $\mathcal{A}(x_i) \subset A$. Any subset of mentions, $e$, also has a corresponding set of attributes, $\mathcal{A}(e) \subset A$, that is derived from its underlying mentions in a process called *canonicalization*. We focus on a simple method of canonicalization, which derives the attributes of a set of mentions, $e$, as the union of attributes of the corresponding mentions. Our model of mentions, entities and attributes is reminiscent of previously proposed Bayesian models used for ER [Steorts et al., 2016].

### 2.2 Hierarchical ER

Like many instances of previous work, we choose to model inferred entities hierarchically [Culotta et al., 2007, Singh et al., 2011, Wick et al., 2012b, Levin et al., 2012, Liu et al., 2014, Zhang et al., 2018]. Hierarchical modeling allows for the usage of a learned entity-level *linkage function* during inference, i.e., a function $g : 2^\mathcal{M} \times 2^\mathcal{M} \to \mathbb{R}$, which scores the compatibility of two sets of mentions (rather than pairs of mentions). Learned linkage functions have been shown to improve entity resolution systems by helping to identify set-level inconsistencies and similarities with respect to attributes like: gender, animacy and number (singular/plural) [Raghunathan et al., 2010, Durrett et al., 2013, Clark and Manning, 2016]. Additionally, hierarchical models promote efficiency in inference and facilitate the representation of uncertainty by encoding multiple partitions of the underlying mentions simultaneously [Zhang et al., 1996, Heller and Ghahramani, 2005, Fichtenberger et al., 2013, Kobren et al., 2017].

We model the set of inferred entities using a binary tree $\mathcal{T}$. Each leaf, $l \in \mathcal{T}$, stores a unique mention, e.g., $l.x = x_i$, and each internal node, $v \in \mathcal{T}$ represents that set of

mentions stored at its descendant leaves, $\text{lvs}(v)$. Each node $v \in \mathcal{T}$ stores an *attribute map*, $m : A \to \mathbb{R}$, that maps attributes to their corresponding *weights*. The attribute map at each leaf, $l.m$, maps all attributes in $l.x$ to 1 as its weight[2]. The attribute map of an internal node, $v.m$, is constructed via canonicalization. For now, consider a canonicalization procedure that constructs that attribute maps of internal nodes as follows:

$$v.m[a] = \sum_{c \in \text{ch}(v)} c.m[a],$$

where $\text{ch}(\cdot)$ returns the children of its arguments, $a$ is an attribute, and $m[a]$ is the weight of $a$ in the map $m$. In words, the weight of an attribute in $v.m$ is the sum of that attribute's weight in $v$'s children's maps. A subset of mentions *exhibits* an attribute $a$ if the weight of $a$ in the corresponding attribute map is greater than 0. Attributes that do not appear in $\mathcal{A}(v)$ have weight 0 in $v.m$.

The compatibility of any two nodes in the tree can be scored via the linkage function, $g$. Each node, $v$, stores its *linkage score*, $v.\sigma$, where the linkage score of a node is computed by evaluating $g$ on the attribute maps of its two children, $\text{ch}(v)$. The linkage score of each leaf is positive infinity. Once the linkage score of all nodes in a tree, $\mathcal{T}$, are computed, the set of inferred entities, $\hat{\mathcal{E}}$, can be extracted from $\mathcal{T}$ using a threshold, $\tau$. In particular, the inferred entities correspond to the tallest nodes in $\mathcal{T}$ whose descendants all possess linkage scores greater than or equal to the threshold.

## 3. User Feedback

Despite significant research effort, ER models are inevitably imperfect and lead to partitions in which mentions from different ground-truth entities are clustered together, or mentions from the same ground-truth entity are split apart. As previously discussed, KB users are well-situated to identify these errors so that the underlying partition of the mentions can be adjusted. However, as with KB mentions, identity uncertainty may permeate user feedback, which must be resolved. In this section, we present a formal model of user feedback that enables joint resolution of identity uncertainty with respect to both mentions and feedback simultaneously.

### 3.1 Feedback Mentions

Recall the example of identity uncertainty shown in Figure 1. The example shows how there can be ambiguity about the entity to which a piece of feedback refers. In the figure's example, the homepage of `Rajarshi Das` is given in *User Feedback #1*. The entity about which the feedback is given is later split, resulting in uncertainty regarding the identity to which it refers.

At a high-level, we propose to represent user feedback as *mentions*. In this way, ER may reason about user feedback and standard mentions jointly. More precisely, each piece of user feedback is represented as a *feedback mention* (FM). Like standard mentions, each FM, $f$, possesses an attribute map, called *packaging*, $f.m_{pack} : A \to \mathbb{R}$ that defines its

---

2. Mapping attributes to weights allows for modeling the *strength* of various attributes. For example, attributes extracted from a data source that is known to be noisy may have lower weight.

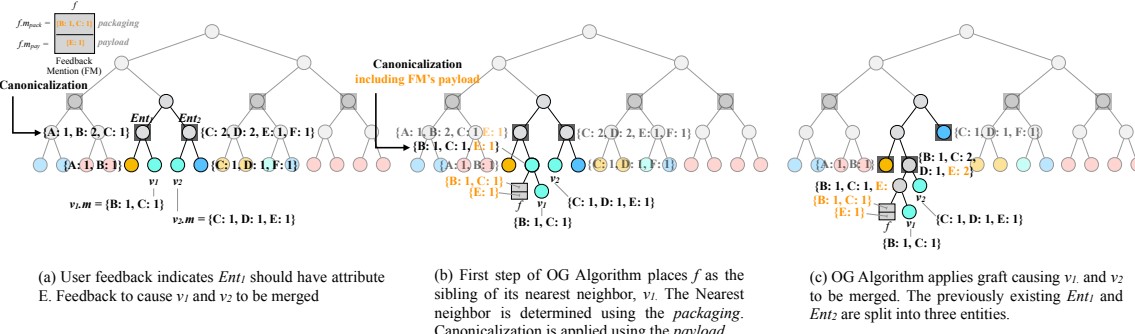

(a) User feedback indicates $Ent_1$ should have attribute E. Feedback to cause $v_1$ and $v_2$ to be merged

(b) First step of OG Algorithm places $f$ as the sibling of its nearest neighbor, $v_1$. The Nearest neighbor is determined using the *packaging*. Canonicalization is applied using the *payload*.

(c) OG Algorithm applies graft causing $v_1$ and $v_2$ to be merged. The previously existing $Ent_1$ and $Ent_2$ are split into three entities.

Figure 2: Example feedback mention (§3.1) added to the KB using the OG algorithm (§4).

compatibility with mentions and other FMs alike via the linkage function. In the context of hierarchical ER (§2.2), this translates to each piece of user feedback being stored at a leaf in the tree, just like standard mentions. ER is run over all mentions and feedback together, which allows the feedback to impact ER. An example FM and its integration in ER, using the OG algorithm (§4), is shown in Figure 2.

## 3.2 Negatively Weighted Attributes

One key difference between FMs and standard mentions is that an FM may be initialized with attributes that map to *negative* weights. Negatively weighted attributes allow feedback to express *incompatibility* with other mentions and FMs. Thus, negatively weighted attributes are used to encourage splits in the underlying partition of the mentions. For example, in the scientific KB discussed above (Section 1), the user supplies feedback claiming that `Rajarshi Das` from UMass Amherst did *not* author the paper *Genetic Reinforcement Learning with Multilayer Neural Networks* from 1991. This feedback can be represented as a mention with name (attribute) "Rajarshi Das" and institution "UMass Amherst" mapped to positive weights and the paper attribute of 1991 paper mapped to a negative weight. The addition of such feedback encourages a split of the inferred KB entity `Rajarshi Das`, which is built from mentions of two, distinct, ground-truth entities as shown in Figure 1.

## 3.3 Correcting Errors from Noisy Mentions

Negatively weighted attributes also allow for the correction of spurious inferred entity attributes that stem from noisy mentions (i.e., mentions with incorrectly extracted attributes). Consider an instance of hierarchal ER in which there exist a node, $v$, with spurious attribute $a$ which stems from a noisy mention, and let $v.f_{\texttt{pack}}[a] = 1$. Since the error stems from noise, the underlying partition of the mentions may not require adjustment; $v$ simply requires the removal of attribute $a$. This can be accomplished with negatively weighted attributes and canonicalization (§2.2). Specifically, consider an FM, $f$, with $f.m_{\texttt{pack}}[a] = -1$. If $f$ were made a descendant of $v$, through canonicalization, the negative weight would be propagated up the tree and $v.m_{\texttt{pack}}[a] = 0$, effectively removing the spurious attribute.

### 3.4 Feedback Mention Payloads

In some cases, like the one described above, a FM may be incompatible with its intended target according to the linkage function (especially when attempting to correct a spurious attribute). Therefore, each FM, $f$, is endowed with a second attribute map, called its *payload*, $f.m_{\texttt{pay}}$. The attributes in a FM's payload are not used by the linkage function to assess its compatibility with other nodes in $\mathcal{T}$. However, payload attributes *do* affect canonicalization. Specifically, for a parent $p$ and children $v$ and $v'$, the packaging at $p$ is computed by summing the weights in the packaging and payload maps of both $v$ and $v'$. This allows attributes in the payloads of $v$ and $v'$ to remain "hidden" from one another until the nodes are merged, at which point these attributes are propagated to their parent and used in subsequent compatibility computations. Using the aforementioned canonicalization notation, the attribute map for the parent node $p$ is:

$$p.m[a] = \sum_{\substack{c \in \texttt{ch}(v), \\ c \text{ is not a FM}}} c.m[a] + \sum_{\substack{c \in \texttt{ch}(v), \\ c \text{ is a FM}}} \left(c.m_{\texttt{pack}}[a] + c.m_{\texttt{pay}}[a]\right)$$

Intuitively, the packaging can be thought of as a set of attributes used to guide initial placement of a FM, while the payload contains missing and incompatible attributes that may negatively affect the FM's placement.

## 4. The Online Grafting Algorithm

Our proposed representation of feedback as mentions is compatible with any hierarchical inference algorithm for ER. In real-world settings, new data and feedback are created and integrated with the KB continuously over time. Thus, we focus on online KB construction, where data points (i.e., mentions and user feedback) are integrated with the KB one at a time.

In this section, we present the online grafting (OG) algorithm for online, hierarchical ER. At all times, the algorithms maintains a tree-consistent partition corresponding to the set of inferred KB entities. This partition is computed via a threshold, $\tau$ (which can be learned at training time). The algorithm is comprised of two recursive subprocedures: swap (`swap_l`) and `graft`, which promote local and global optimality, respectively. The OG algorithm proceeds as follows. When a new data point, $x$, arrives, it is added as a sibling of its nearest neighbor leaf, $v$. Note that adding $x$ as a sibling of $v$ makes $v$'s previous sibling, $a$, into the aunt of both $x$ and $v$. Next, we invoke the `swap_l` subroutine, recursively. During a `swap_l`, consider $x$, $v$ and $a$ and check whether

$$g(v, x) < g(v, a),$$

i.e., whether the $v$ and it's previous sibling, $a$, are more compatible than $v$ and $x$. If so, swap the positions of the two subtrees rooted at $x$ and $a$ (the result of which has $a$ and $v$ as siblings and $x$ as their aunt). Repeat this procedure until $x$'s sibling is more compatible with $x$ than with its previous sibling, or until $x$'s sibling is the root of an inferred entity. See Algorithms 1 and 2 for pseudocode.

After swapping terminates, the `graft` subroutine is invoked recursively from the parent of $x$, $\texttt{par}(x) = p$. A `graft` invoked from a node whose linkage score is below the threshold,

---

**Algorithm 1** Insert$(x, \mathcal{T}, g)$

---

$v = $ nearestNeighbor$(x, g, \mathcal{T})$
split_down$(v, x)$
$a = $ aunt$(v)$
**while** swap_l$(\mathcal{T}, g, x, v, a)$ & $v.\sigma > \tau)$ **do**
  $v = $ sibling$(x)$
  $a = $ aunt$(v)$
graft$(\mathcal{T}, g, $ par$(x))$

---

**Algorithm 2** swap_l$(\mathcal{T}, g, x, v, a)$

---

**if** $g(v, x) < g(v, a)$ **then**
  Swap $x$ and $a$ in $\mathcal{T}$
  **return true**
**else**
  **return false**

---

**Algorithm 3** graft$(\mathcal{T}, g, p)$

---

**while** $p.\sigma > \tau$ **do**
  $v' = \text{argmax}_{\ell \in \text{lvs}(\mathcal{T}) \setminus \text{lvs}(p)} \, g(p, \ell)$
  **if** $g(p, v') > \max\{g(p, \text{sib}(p)), g(v', \text{sib}(v')), \tau\}$
  **then**
    Move $v'$ to be the sibling of $p$
    p $= $ par(p)
  **else**
    **break**

---

Figure 3: Online Grafting Algorithm.

$\tau$, terminates immediately and no further grafts are attempted. If $p.\sigma > \tau$, search the leaves of $\mathcal{T}$ for, $v'$, the most compatible leaf with $p$ that is not a descendant of $p$. Test whether,

$$g(p, v') > \max\{g(p, \text{sib}(p)), g(v', \text{sib}(v')), \tau\}$$

i.e., $p$ and $v'$ are more compatible with each other than with their respective siblings, and their linkage score is higher than the threshold $\tau$. If the test succeeds, make $v'$ the sibling of $p$ and re-invoke the graft subroutine from par$(p)$. If the test fails, consider 3 cases:

1. if $g(p, v') \leq \tau$ then re-invoke the graft subroutine from par$(p)$;

2. if $g(p, \text{sib}(p)) > g(p, v')$ then repeat the test between par$(p)$ and $v'$;

3. if $g(v', \text{sib}(v')) > g(p, v')$, then repeat the test between $p$ and par$(v')$.

Intuitively, the graft subroutine iteratively attempts mergers between ancestors of $x$ and nodes compatible with those ancestors in $\mathcal{T}$. Notice that a merger between two nodes in $\mathcal{T}$ can only occur if: 1) both nodes are more compatible with each other than with their siblings and 2) their resultant parent has a linkage score higher than the threshold, i.e., the two nodes belong to the same inferred entity. The graft subroutine promotes global optimality and helps to make ER more robust to data point arrival order. See Algorithm 3 for pseudocode and Figure 4 for an illustration of the algorithm's tree operations.

## 5. Training

Recall that the linkage function, $g$, takes two nodes, $v, v' \in \mathcal{T}$ and returns their compatibility. Define the precision of a node pair $(v, v')$ to be

$$\text{pre}(v, v') = \frac{|\{(l, l') : l \in \text{lvs}(v), l' \in \text{lvs}(v'), e^\star(l) = e^\star(l')\}|}{|\text{lvs}(v)| \times |\text{lvs}(v')|}.$$

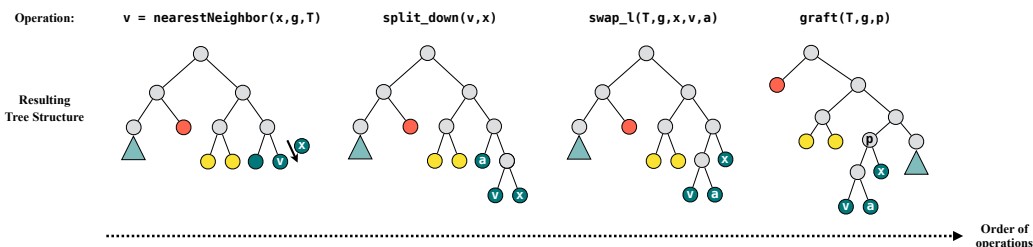

Figure 4: Online Grafting (OG) Algorithm's tree re-arrangement operations applied sequentially with the additional of a new mention, labeled $x$.

In words, the precision of a pair is the fraction of leaf-pairs, where one leaf is a descendent of $v$ and the other a descendent of $v'$, where both leaves belong to the same ground-truth entity. Note that FMs have no ground-truth label and are not included in this calculation. We train $g$ to regress to the precision of its input node pair. Our training procedure is inspired by work in entity and event coreference that trains a linkage function to regress to precision of mergers in the context of bottom up agglomerative clustering [Lee et al., 2012].

We select nodes for training in the context of running the OG algorithm. When a new mention, $x$, arrives, we generate training examples between it and all other leaves (mimicking the nearest neighbor search). Next, we generate training examples between $x$ and ancestors of its nearest neighbor (according to model score) resembling the swap_l subroutine. Finally, we insert $x$ into the tree and generate training examples between the parent of $x$ (from which the graft subroutine is initiated) and all nodes that can be viably grafted. The same procedure is repeated for each incoming mention.

The linkage function, $g$, also needs to be trained to appropriately handle user feedback. After a batch of $N$ training mentions have been added to a tree, we generate up to $N$ pieces of user feedback (generated via the DETAILED scheme, discussed in Section 6). The generated feedback is either *positive* and intended to encourage a graft, or *negative* and intended to encourage a split of some inferred entity. The training feedback are also inserted into the tree, one at a time, using the OG algorithm. For each piece of feedback, we generate a training example between the feedback and its intended sibling in the tree (designated at feedback generation time), and set the precision of the example to be 1.0. Next, a merge between the feedback and its intended sibling is hallucinated and a training example is generated between the resulting parent node and the feedback's *target*. For positive feedback, the target is the root of a subtree that belongs near the feedback's sibling in the tree, and for negative feedback, the target is a node with which the feedback and its sibling are incompatible. These two types of examples help to train the model to use positive feedback to encourage grafting and negative feedback to encourage splitting.

We tune the threshold, $\tau$, on a development set. In particular, at regular intervals throughout training, a hierarchical clustering of a set of development mentions is constructed using the OG algorithm and the current linkage function model parameters. A search is performed measuring the pairwise F1 score[3] for the selection of entities determined by each

---

3. Precision/Recall/F1 are computed on the pairs of mentions that clustered together by the algorithm compared to the ground truth. For more information, we refer the reader to Menestrina et al. [2010]

value of $\tau$. At test time, the parameters and threshold that resulted in the best partition of the hierarchy on the dev set are used.

## 6. Experiments

We perform experiments testing various styles feedback representation in the context of online *author disambiguation*—a particular instantiation of ER in which the ground-truth entities are real-world authors. We use the Rexa author disambiguation dataset, which includes 8 author *canopies*; each canopy contains ambiguous mentions of authors with the same first initial and last name [Culotta et al., 2007]. The mentions are derived from publications and contain: coauthors, titles, publishing venue, and year of publication. The goal is to partition the mentions by real-world author.

### 6.1 Setup

Our experimental setup is composed of two phases. In the first phase, the set of mentions arrive online and are incrementally added to a hierarchical clustering, $\mathcal{T}$, using the OG algorithm (§4). The second phase proceeds in rounds. At the start of round $t$, a set of inferred entities, $\hat{\mathcal{E}}_t$, is constructed using a threshold, $\tau$, tuned at training time on a development set (§5). If $\hat{\mathcal{E}}_t = \mathcal{E}^\star$, then the episode terminates. Otherwise, we simulate user interaction by generating feedback, $f_t$, made with respect to a randomly selected inferred entity, $\hat{e} \in \hat{\mathcal{E}}_t$. Then, $f_t$ is added to $\mathcal{T}$ using the OG algorithm, potentially triggering a repartitioning of the mentions. No more than 400 rounds are permitted. Although rare, if after 400 rounds, the ground-truth entities have not been discovered, the method is recorded as having taken $400 + d$ rounds, where $d$ is the number of mentions that would need to be swapped to discover $\mathcal{E}^\star$. We measure the mean number of rounds required to discover $\mathcal{E}^\star$ for each method, repeated over 25 trials, and report a paired-$t$ statistic (and corresponding significance level) between each baseline feedback representation style and our proposed FM representation.

### 6.2 Simulating Feedback

We simulate *positive* and *negative* feedback using node *purity* and *completeness*. A node $v \in \mathcal{T}$ is *pure* if:

$$\exists i \ \text{ s.t. } \ \forall l \in \texttt{lvs}(v), \ e^\star(l) = e_i^\star,$$

i.e., all of mentions stored at the leaves of $v$ correspond to the same ground-truth entity, $e_i^\star$. A node $v \in \mathcal{T}$ is *complete* if:

$$\exists i \ \text{ s.t. } \ \{l \in \texttt{lvs}(\mathcal{T}) : e^\star(l) = e_i^\star\} \subseteq \texttt{lvs}(v'),$$

i.e., that $v$'s leaves contain *all* mentions of some ground-truth entity $e_i^\star$.

To generate both positive and negative feedback, we sample an intended *destination* and an intended *target*. The destination is a particular node in the tree to which the feedback is intended to be similar. The target of the feedback is a different node that the feedback is intended to be merged with or separated from upon insertion. Note that even with full

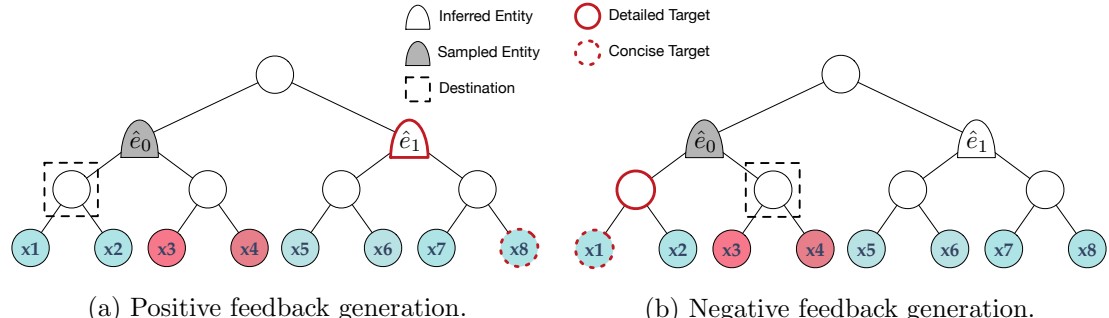

(a) Positive feedback generation.  (b) Negative feedback generation.

Figure 5: **Detailed and Concise feedback.** To generate either positive or negative feedback, begin by randomly sampling an inferred entity. Then, sample a *destination*–the root of a pure subtree that is also a descendant of the sampled entity. The packaging of the feedback contains the attributes at the destination. Finally, sample a *target*, which is used to construct the feedback's payload. The target is a sampled mention in the CONCISE setting, or the largest, pure ancestor of a sampled mention in the DETAILED setting.

knowledge of the destination and the target (and the ER algorithm), it is difficult to design feedback that will cause the intended tree rearrangements exactly because *other* nodes in $\mathcal{T}$ may interfere during nearest neighbor search by being very compatible with either the feedback, target, or destination.

### 6.2.1 DETAILED AND CONCISE POSITIVE FEEDBACK

Positive feedback is constructed with the intention of merging two nodes in the tree via a `graft`. To generate positive feedback, sample a node that is the root of a pure and incomplete subtree and whose parent is impure, $r$; this node is the destination of the feedback. Then, randomly select a mention, $x$, that is of the same ground-truth entity as the leaves of $r$, but is not a descendant of $r$. If constructing CONCISE feedback, $x$ is the target of the feedback; if constructing DETAILED feedback, traverse the ancestors of $x$ until $s$, the first ancestor of $x$ whose parent is impure. The node $s$ becomes the target of the feedback. See Figure 5a for a visual illustration.

### 6.2.2 DETAILED AND CONCISE NEGATIVE FEEDBACK

Negative feedback is constructed with the intention of splitting an inferred entity. We simulate negative feedback by randomly sampling an impure inferred entity (i.e., subtree) and finding its root, $r'$. We construct the destination of the feedback by randomly sampling a mention $x' \in \text{lvs}(r')$ and finding $s'$, the ancestor of $x'$ closest to the root of $\mathcal{T}$ that is pure. If constructing CONCISE feedback, sample a mention $x'' \in \text{lvs}(r') \setminus \text{lvs}(s')$ to be the target; if constructing DETAILED feedback, traverse the ancestors of $x''$ until $s''$, the ancestor of $x''$ closest to the root of $\mathcal{T}$ that is pure. In both cases, $s''$ becomes the target of the feedback. See Figure 5b for a visual illustration.

### 6.3 Baseline Methods

Integrating user feedback under identity uncertainty has not been the subject of significant study. Therefore, we propose and compare the following baseline feedback representations:

1. **Feedback Mentions (FM)** - the approach advocated in this work. Feedback is constructed with packaging and payload attribute maps and is integrated with existing mentions via the OG algorithm.

2. **Packaging Mentions (pack)** - similar to our approach, but the feedback mentions have no payloads. All attributes that would have been included in a payload are instead added to the corresponding packaging.

3. **Hard Assignment (assign)** - generate feedback with both packaging and payload. But then, find the node $v \in \mathcal{T}$ to which it would have been made a sibling by the OG algorithm and *permanently* assign the feedback to $v$. If $v$ is an internal node and any of its leaves are ever moved (e.g., by a `graft`) such that they are no longer descendents of $v$, remove and delete all feedback assigned to $v$, since, because of identity uncertainty, it is unclear whether the feedback was intended to apply to the moved mentions.

4. **Hard Mention Assignment (assign-m)** - similar to the assign approach but the feedback must be assigned to a *leaf* in $\mathcal{T}$. Since mentions are atomic (rather than ephemeral, like inferred entities), the assigned feedback is never deleted.

### 6.4 User Feedback about Author Expertise

Our first experiment resembles a scenario in which users interact with a KB of scientists and provide feedback with respect to the KB's belief about a scientist's expertise. The expertise of an inferred entity is represented as a bag of key phrases drawn from the titles of its underlying mentions. Users supply missing keywords and identify incorrect keywords. In this experiment, packaging contains the set of attributes at the sampled destination and the payload contains the keywords at the target (generated from mention titles). Importantly, expertise key phrases are a *shared* attribute, that is, multiple ground-truth entities exhibit some of the same expertise. Example Rexa data and simulated user feedback is shown in 6

### 6.5 Authorship Feedback

Our second experiment resembles the scenario in which a user browses a KB of scientist profiles, similar to Google Scholar[4], and identifies incorrectly assigned and missing publications. Similar to the first experiment, the packaging contains attributes mapping to positive weights in the sampled destination. However, here, payloads contain titles stored in the sampled targets. Note that publication authorship *is not* a shared attribute, i.e., no two ground-truth entities in the same canopy have collaborated on any publication.

### 6.6 Results and Discussion

Tables 1a and 1b contain the results of the expertise and title experiments, respectively. Each table reports the paired $t$-statistic between each baseline method and our proposed

---

4. https://scholar.google.com/

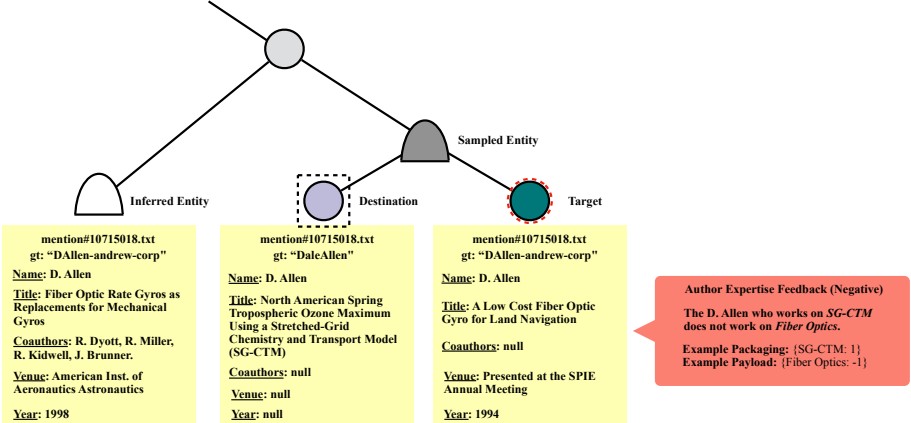

Figure 6: Example Mentions from the Rexa dataset as three leaves in a tree created by OG. Example feedback is shown for a concise target. The packaging and payload for authorship FM would contain the two titles mentioned with positive and negative weights respectively.

approach (FM), under DETAILED and CONCISE feedback generation schemes, with respect to the number of pieces of feedback required to discover the ground-truth partition of the mentions. Each row represents a canopy in which the experiment is performed, and each column corresponds to a baseline method and feedback generation setting. Each cell contains the difference between the mean number of rounds required by the FM approach and a baseline approach to discover the ground-truth partition (higher is better). Positive numbers are bolded; asterisks (*) indicate statistical significance ($p < 0.05$) and two asterisks (**) indicate statistical significance ($p < 0.01$). Rows are omitted if the initial partition of the mentions, constructed by the OG algorithm and subject to no user feedback, is correct.

The paired-$t$ statistics compare our proposed feedback representation (FM) to the three baseline feedback representations. We find that **FM** outperforms **pack** in both the DE-TAILED and CONCISE settings of Experiment I on all but two of the canopies. In 7 out of 14 canopies, the results are statistically significant. These results underscore the importance of using only certain attributes (stored in the packaging) during the initial nearest neighbor search. We hypothesize that storing shared attributes in the payload is especially important because otherwise they can interfere with initial routing. When feedback is made with respect to attributes that are not shared, as in Experiment II, separating packaging and payload is less important. This is evidenced by the **pack** approach slightly outperforming **FM**s in the detailed setting, but never significantly. **FM**s generally outperform **pack** in the CONCISE setting. We hypothesize that this is a result of better initial placement of the feedback in the tree by the OG algorithm.

In comparing, **FM** and **assign** we find that our proposed approach typically performs better in Experiment II while the baseline performs better in Experiment I. We note that the feedback in Experiment I is more ambiguous than Experiment II (because expertise is a shared attribute). We hypothesize that **assign**'s better performance in Experiment I is due to the baseline's approach of *deleting* feedback to mitigate errors caused by identity uncertainty with respect to user feedback. We note that this agrees with the observation

| Canopy | DETAILED | | | CONCISE | | |
| | vs. **assign** | vs. **assign-m** | vs. **pack** | vs. **assign** | vs. **assign-m** | vs. **pack** |
| --- | --- | --- | --- | --- | --- | --- |
| allen_d | $-2.07^*$ | **4.03**$^{**}$ | **3.41**$^{**}$ | **1.88** | **2.20**$^*$ | **0.75** |
| blum_a | $-1.18$ | **1.59** | **1.27** | $-0.85$ | **0.20** | **0.61** |
| jones_s | **1.37** | **7.74**$^{**}$ | **6.67**$^{**}$ | **1.11** | **0.65** | **4.04**$^{**}$ |
| lee_l | $-1.53$ | **1.02** | **4.04**$^{**}$ | $-0.22$ | $-0.28$ | **0.95** |
| moore_a | **0.90** | **3.25**$^{**}$ | **3.66**$^{**}$ | $-0.30$ | $-0.71$ | $-0.49$ |
| robinson_h | $-0.99$ | $-3.60^{**}$ | $-0.81$ | $-0.76$ | $-0.61$ | **0.27** |
| young_s | $-0.71$ | **1.09** | **4.87**$^{**}$ | **0.76** | **0.45** | **2.17**$^*$ |

(a) Experiment I: keyword feedback.

| Canopy | DETAILED | | | CONCISE | | |
| | vs. **assign** | vs. **assign-m** | vs. **pack** | vs. **assign** | vs. **assign-m** | vs. **pack** |
| --- | --- | --- | --- | --- | --- | --- |
| allen_d | **4.82**$^{**}$ | **2.67**$^*$ | $-1.96$ | **5.86**$^{**}$ | **2.01** | **1.98** |
| blum_a | **0.11** | $-0.66$ | $-0.98$ | **0.98** | **1.03** | **5.70**$^{**}$ |
| jones_s | **1.49** | **1.99** | **0.85** | **1.00** | **1.02** | **3.03**$^{**}$ |
| lee_l | **0.64** | **1.06** | $-1.03$ | $-1.44$ | **0.47** | $-0.58$ |
| moore_a | **1.30** | **1.97** | $-0.90$ | **1.94** | **1.09** | **1.05** |
| young_s | $-1.87$ | **0.76** | $-1.88$ | **0.93** | **0.91** | **0.97** |

(b) Experiment II: title feedback.

Table 1: **Paired-t statistic.** Each cell represents that difference in mean number of feedback-rounds required to discover the ground-truth entities over 25 runs between a baseline, denoted by the column heading, and our proposed approach (FM). Positive numbers indicate that FM requires *fewer* rounds of feedback than its competitor (larger numbers are better). Two asterisks (**) indicates that the statistic is significant at a 0.01 significance level; one asterisk indicates statistical significance at the 0.05 level. The `mcguire_j` canopy is excluded from Tables 1a and 1b and the `robinson_h` canopy is excluded from Table 1b since in these canopies, either: 0 or 1 edits are required to discover the ground-truth entities across baselines.

that **FM** generally outperforms **assign-m** in both experiments, in that **assign-m** is similar to the **assign** strategy but never deletes feedback.

## 7. Related Work

Effective utilization of user feedback has been the subject of significant study in the context of KB construction. Early work, like NELL, primarily enlists humans for labeling data, which are used to train downstream models [Carlson et al., 2010]. Other work has used active learning in training relation extraction models [Angeli et al., 2014b]. Another approach employed by the DeepDive system asks humans to identify relevant features by writing feature extraction rules in support of KB construction [Ré et al., 2014, Angeli et al., 2014a].

The study of leveraging user feedback in ER has primarily focused on the solicitation of pairwise feedback. For example, given a set of mention pairs, the CrowdER system automatically prunes the set of pairs that are highly unlikely to be coreferent, and then constructs

crowdsourcing HITs to collect binary labels for the remaining pairs [Wang et al., 2012]. In other similar work, human are asked to identify matching mentions across databases in data integration [Li, 2017]. Recent work studies online ER with an oracle, in which the goal is to design efficient strategies for soliciting humans for pairwise constraints among mentions [Vesdapunt et al., 2014, Mazumdar and Saha, 2016, 2017].

Recent work in author coreference also involves humans-in-the-loop [Zhang et al., 2018]. This work discusses both pairwise constraints as well as *identity constraints*. Unlike our work, their identity-level feedback is treated as a collection of pairwise constraints. As we point out, feedback that can be reduced to a set of pairwise constraints is insufficient for general KB feedback as pairwise feedback is only designed for correcting errors in ER (and not general KB errors). Similarly, many examples of user feedback are inexpressible using pairwise constraints.

The OG algorithm is closely related to the recently proposed clustering algorithm, GRINCH [Monath et al., 2019], which also uses a graft procedure in an incrementally built hierarchical clustering. Unlike GRINCH, OG uses a threshold $\tau$ to determine when tree re-arrangements are made and to maintain the current set of inferred entities.

The most closely related work to ours is a preliminary study of incorporating user feedback in the context of data integration [Wick et al., 2013, 2012a]. In this work, users supply pairs of mention-like records that posses either should-link or should-not-link factors, which either softly repel the pair or encourages their merger.

## 8. Conclusion

This work presents a framework for reasoning about user feedback under identity uncertainty during KB construction. We advocate representing user feedback as feedback mentions that participate in ER alongside standard mentions. Our feedback mentions are endowed with a packaging–used to identify similar mentions during ER–and a payload–that is used to add missing attributes to inferred entities, correct mistakes and influence future ER decision. We give a hierarchical model of inferred entities and present the OG algorithm for performing online ER amongst standard and feedback mentions. In experiments, we show that our approach often outperforms baseline approaches in terms of efficiency with respect to recovering the ground-truth partition in ER. Our work is a foundational step in addressing a significant and under-explored problem in automatic KB construction whose solution could improve the accuracy and efficacy of integrating expressive user feedback with KB content.

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
