# OpenReview forum: "Integrating User Feedback under Identity Uncertainty in Knowledge Base Construction"
_AKBC.ws/2019/Conference — AKBC 2019_

### Official Review · AnonReviewer1 · 2019-01-09
**A nice algorithmic contribution for integrating user feedback for KBC**

**Rating:** 7
**Confidence:** 3

**Review:**

This paper introduces a method to integrate user feedback into KBs in the presence of identity uncertainty, a problem that arises in the integration of new data in knowledge base construction. The proposed method represents user feedback as feedback mentions and uses an online algorithm for integrating these mentions into the KB.

The paper targets an important problem in knowledge base construction, i.e., integrating user feedback in the online setting. The proposed hierarchical model looks reasonable and effective. And overall, the work is well presented.

The paper makes an algorithmic contribution. The contribution is, however, limited from the perspective of human computation. The experiment uses simulated user feedback for evaluating the method. In real-world settings, user feedback can be skewed to certain types (e.g., negative feedback) or be noisy (so the feedback is not reliable). How would these affect the result?

---

> ### Author Response · Authors · 2019-01-20
> **Experiment with mixed detailed and concise feedback**
>
> Thank you for your positive review!
> We would be happy to expand on our experimental results. While it is not possible for us to carry out a study with real human feedback (which requires IRB approval, etc.) we can perform experiments with a mixture of detailed and concise feedback and with different proportions of positive and negative feedback. Would this address your criticism (at least in part)?

---

### Official Review · AnonReviewer2 · 2019-01-11
**lack of details, few novelty**

**Rating:** 3
**Confidence:** 4

**Review:**

This paper presents a hierarchical framework for integrating user feedback for KB construction under identity uncertainty.

1. it is unclear about the algorithm implementation, such as what is the implementation of feedback mention.

2. There is a definition about attribute map, but I can't find where the model uses it. Same thing for the precision of a node pair.

3. How to calculate the function g(.) is unclear either.

4. In section 5.2, the COMPLETE definition seems not correct, it is still the definition of PURE.

5. The example for constructing positve/negative feedback is too vague.

6. The experiment section needs more analysis including qualitative result.

---

> ### Author Response · Authors · 2019-01-20
> **Details are included in the paper**
>
> Thank you for your review.
> The details you mention do appear in our paper, but perhaps they can be further clarified.  We address each of your criticisms below:
>
> 1. The implementation of feedback mentions is described in the third paragraph of section 2.3: “FMs are composed of two attribute maps” the first called “packaging” and the second called “payload”.  Immediately after this description, we provide intuition for why both attribute maps are needed. In section 3.1.1, we describe precisely how both of these attribute maps are used during canonicalization.
>
> 2. The model, g, computes the linkage score between two nodes using their attribute maps.  This is described in the second paragraph in 3.1.1: “Since g scores the compatibility of two nodes based on their attribute maps...”. Perhaps we can further clarify this in the paper by including some text describing the linkage function g more precisely.
>
> The precision of a node pair is used during training. In the first paragraph of section 4: “We train g to regress to the precision…” Thus, during learning we run agglomerative clustering and, at each merge, train g to predict the precision of the corresponding node pair.
>
> 3. The function g is computed from the attribute maps of the corresponding nodes (which differ by dataset).  It is true that we omit the specific features that we use, for example, whether two nodes have the same title with a corresponding positive count in their respective attribute maps, or whether one node contains an attribute A with a positive count and the second node contains the attribute A with a negative count.  We are open to including these features in a supplement.
>
> 4. We agree that the definition for complete could be revised to be clearer. We will modify the definition as follows:
> A node v is complete if for some $i$,
> \lvs{e^\star_i} \subseteq lvs{v}.
> Thus, if for some $i$, the leaves of $e^\star_i$ are a subset of the leaves of $v$, then $v$ is complete. We note that the definition for complete is described in words immediately after the mathematical statement in the paper and is different from the definition of pure.
>
> 5. What part of the detailed/concise feedback needs further explanation? As described in 5.2.1 and 5.2.2 and as detailed in the figure, the simulation procedure finds a “destination” and “target” for each edit.  The simulated edit contains a packaging that includes all of the positive attributes at the destination and a payload containing the attributes at the target (either with positive or negative weight, as described in the paper). We’re happy to describe the feedback generation process in more detail.
>
> 6. We believe that qualitative results would not be very illuminating, but we’re open to including some. Would you find informative an  example of an edit added to a tree and the subsequent modification of inferred entities?

---

### Official Review · AnonReviewer3 · 2019-01-30
**Relevant contribution for humans-in-the-loop in KB construction**

**Rating:** 7
**Confidence:** 4

**Review:**

The paper presents a novel solution to an interesting problem - when KBs are automatically expanded user feedback is crucial to identify incorrect and missing entity attributes and relations. More specifically, in the case of entity identity uncertainty, enabling the user feedback as part of the entity resolution mentions, appears novel and important.
The paper is well written and organized.

Points for improvement:
- It would be interesting to see more in-depth analysis on examples where the proposed approach fails, and based on this to also outline open issues and future work.
- The human computation aspects of the paper are lacking sufficient explanation in terms of implementation in real settings, as well as positioning with related work in human computation research
- it would be interesting to know, what is the experimental design that authors would consider for an evaluation with actual user feedback vs. the simulated one

---

### Meta-Review · Area_Chair1 · 2019-02-12
**Interesting topic to be presented**

**Recommendation:** Accept (Poster)
**Confidence:** 4

**Metareview:**

The paper presents an interesting methodology. The results are interesting, however the paper really misses out on an in-depth discussion and reflection of the pros and cons of this approach as well as on a proper related work comparison to similar approaches.

---

### Decision · Program_Chairs · 2019-02-15
**AKBC 2019 Conference Decision**

Accept